# Cyclic Peptide Mimotopes for the Detection of Serum Anti–ATIC Autoantibody Biomarker in Hepato-Cellular Carcinoma

**DOI:** 10.3390/ijms21249718

**Published:** 2020-12-19

**Authors:** Chang-Kyu Heo, Hai-Min Hwang, Won-Hee Lim, Hye-Jung Lee, Jong-Shin Yoo, Kook-Jin Lim, Eun-Wie Cho

**Affiliations:** 1Rare Disease Research Center, Korea Research Institute of Bioscience and Biotechnology, 125 Gwahak-ro, Yuseong-gu, Daejeon 34141, Korea; ckheo40@kribb.re.kr (C.-K.H.); purity-32@hanmail.net (H.-M.H.); wonhee@kribb.re.kr (W.-H.L.); 2Department of Functional Genomics, University of Science and Technology, 125 Gwahak-ro, Yuseong-gu, Daejeon 34141, Korea; 3ProteomeTech Inc. 401, Yangcheon-ro, Gangseo-gu, Seoul 07528, Korea; hj0808@proteometech.com (H.-J.L.); kjlim@proteometech.com (K.-J.L.); 4Biomedical Omics Group, Korea Basic Science Institute, 162 YeonGuDanji-Ro, Ochang-eup, Cheongju, Chungbuk 28119, Korea; jongshin@kbsi.re.kr

**Keywords:** autoantibody biomarker, ATIC, hepatocellular carcinoma, cyclic peptide mimotope, human serum ELISA

## Abstract

Tumor-associated (TA) autoantibodies have been identified at the early tumor stage before developing clinical symptoms, which holds hope for early cancer diagnosis. We identified a TA autoantibody from HBx-transgenic (HBx-tg) hepatocellular carcinoma (HCC) model mouse, characterized its target antigen, and examined its relationship to human HCC. The mimotopes corresponding to the antigenic epitope of TA autoantibody were screened from a random cyclic peptide library and used for the detection of serum TA autoantibody. The target antigen of the TA autoantibody was identified as an oncogenic bi-functional purine biosynthesis protein, ATIC. It was upregulated in liver cancer tissues of HBx-tg mouse as well as human HCC tissues. Over-expressed ATIC was also secreted extracellularly via the cancer-derived exosomes, which might cause auto-immune responses. The cyclic peptide mimotope with a high affinity to anti-ATIC autoantibody, CLPSWFHRC, distinguishes between serum samples from HCC patients and healthy subjects with 70.83% sensitivity, 90.68% specificity (AUC = 0.87). However, the recombinant human ATIC protein showed a low affinity to anti-ATIC autoantibody, which may be incompatible as a capture antigen for serum TA autoantibody. This study indicates that anti-ATIC autoantibody can be a potential HCC-associated serum biomarker and suggests that autoantibody biomarker’s efficiency can be improved by using antigenic mimicry to native antigens present in vivo.

## 1. Introduction

Cancer is the second leading cause of death globally and the most critical barrier to increasing life expectancy in every country of the world in the 21st century [1]. However, extensive studies on cancer and the development of effective therapies during the last decade of genomic and proteomic research make cancer a more preventable and curable disease [2]. Notably, general awareness of the risk factors for cancer and early cancer diagnosis have improved survival. Therefore, it is crucial and urgent to identify more effective and less invasive tumor markers, guiding early diagnosis and therapeutic strategies for individual patients.

Cancer cells can induce immunological responses [3]. The various abnormal substances produced by cancer cells can be antigenic, which are no longer recognized as self. The immune system responds to these non-self antigens, producing specific autoantibodies. Tumor-associated (TA) autoantibodies have been observed in patients with various tumors, including breast [4], prostate [5,6], lung [7], colorectal [8], ovarian [9,10], and liver [11], and have become of interest as cancer biomarkers because they can be easily detected in serum via minimally invasive blood collection [3].

Of course, antigens released by tumor cells, called TA antigens, have been used as the most common tumor biomarkers [12,13,14]. Recently, tumor-secreted exosomes are also suggested as promising cancer biomarkers [15]. They are regarded as real-time monitoring biomarkers of cancer, and enormous efforts have been performed to explain their relationships to tumorigenesis. However, the amounts of TA antigens or TA exosomes in serum depend on the size of the tumor burden, limiting their detection in the early stage of disease despite using highly sensitive devices. On the contrary, TA antibodies are measurable even several years before developing clinical symptoms [3,16]. An anti-p53 autoantibody is the first historical report of early diagnostic TA autoantibody marker, which was detected as early as 17–47 months before clinical tumor manifestation in uranium workers at high risk of lung cancer development [3,17]. Additionally, TA autoantibodies are more stable in vitro than other protein biomarkers, which facilitates their detection. Because of these advantages, TA autoantibodies are emerging as strong candidates for clinically useful cancer biomarkers [18]. However, TA autoantibody’s diagnostic efficacy is often very low, limiting the development of relevant diagnostics.

In this study, we have identified an anti-ATIC autoantibody as a novel autoantibody biomarker for liver cancer. HBx-transgenic HCC-model mouse [19,20] was used to obtain B cell hybridomas producing TA autoantibodies, and a monoclonal TA autoantibody, XC154, was purified and characterized. Its specific TA antigen was identified as ATIC, a purine synthesis protein overexpressed in human tumor cells. Moreover, the mimotopes specific for XC154 autoantibody were screened from a random cyclic peptide library, and the highly reactive mimotope against XC154 autoantibody was proved to be useful for the diagnosis of liver cancer. The reactivity of the recombinant human ATIC (hATIC) expressed in *E. coli* against XC154 autoantibody was also tested; however, the binding of hATIC to XC154 autoantibody was relatively low, not being suitable for the development of in-vitro diagnostics. Based on these results, we discuss the usage of anti-ATIC autoantibody biomarker for cancer diagnosis and the suitable TA epitopes for the development of autoantibody biomarker detection.

## 2. Results

### 2.1. A Tumor-Associated Autoantibody in the HBx-tg HCC Model Mouse Showed the Elevation of Its Target Antigen in Human HCC Tissues

H-*ras12V* transgenic or HBx-transgenic mice have proven to be suitable for the human hepatocellular carcinoma (HCC) model [20,21]. These tumor model mice spontaneously generated liver cancer about 6~10 months after birth. We constructed a B-cell hybridoma pool using tumor-bearing transgenic mice, enriched with B cell hybridomas producing HCC-associated autoantibodies. Several TA autoantibodies from these B cell hybridoma cells have been characterized and proposed as biomarkers for cancer diagnosis [19,22,23,24]. In this study, another monoclonal TA autoantibody, named XC154 mAb, which bound specifically to human tumor cells, was purified, and its antigenic characteristics were characterized.

XC154 mAb was IgM isotype, which was confirmed by antibody isotyping and SDS-PAGE (Figure 1A). It has reacted with a specific antigen (named XC154 Ag) with a molecular weight of about 60 kDa in liver cancer tissues of H-*ras12V* transgenic mice. It also detected the same antigen in non-transgenic mice; however, its expression was higher in tumor tissues about three-fold (*p* < 0.0001; Figure 1B), which shows that the overexpression of XC154 antigen is related to tumorigenesis. XC154 antigen was also expressed ubiquitously in various human tumor cells, including hepatocellular carcinoma (HepG2, Hep3B, Huh7, SK-Hep1), lung cancer (A549), and breast cancer (SK-BR-3, MCF7), as shown by Western blot and immunofluorescence analysis (Figure 1C,D). Immunohistochemical staining with XC154mAb confirmed the elevation of XC154 antigen in human HCC tissues compared to non-neoplastic liver tissues (*p* < 0.05; Figure 1E). Collectively, XC154 tumor-associated autoantibody, which was identified from the mouse model of HCC, detected tumorigenic antigen in the HCC-model mice and human tumors.

### 2.2. The Target Antigen of Tumor-Associated Autoantibody XC154 Was Identified as ATIC, a Bifunctional Purine Biosynthesis Protein

To identify the target antigen of XC154 mAb, we performed mass spectrometric (MS) analysis of the XC154 antigen. The lysate of LNCap/LN3 cells was fractionated by ion exchanger chromatography, and the fractions were analyzed by Western blotting with XC154 mAb. The fractions enriched with the XC154 target antigen (Appendix A) were pooled and separated on the preparative 10% SDS-PAGE gel. The XC154 antigen band corresponding to the immuno-stained antigen was excised and analyzed by tandem mass spectroscopy (Figure 2A).

Ten protein candidates were identified, as shown in Table 1. ATIC, a bi-functional enzyme that possesses the last two activities in *de novo* purine biosynthesis, showed the highest protein score and the number of peptides matched. It also showed a predicted molecular weight of 64 kDa, similar to that of the XC154 antigen. Therefore, we assumed the XC154 antigen as ATIC and validated it further. The knockdown of ATIC using siRNA suppressed the expression of the XC154 antigen (Figure 2B). Three siRNA specific to ATIC suppressed the expression of ATIC transcript in HepG2 cells, and these cells showed the decreased expression of the XC154 antigen. We also analyzed the immunoprecipitates obtained with anti-ATIC antibody by Western blotting using XC154 mAb, or vice versa (Figure 2C), which confirmed that the XC154 antigen which induced the expression of specific TA autoantibody in HCC-model mouse is ATIC. The expression of ATIC in liver tissues of the HCC model mouse was analyzed to validate ATIC as a tumor-associated antigen. HCC tissues from H-*ras12V*-Tg mice showed the elevated expression of ATIC in Western blot analysis (Figure 2D). These results were consistent with the expression pattern of XC154 antigen in HCC tissues from H-*ras12V*-Tg mice probed with XC154 mAb (Figure 1B).

Immunohistochemical analysis of HCC tissues from HBx-Tg as well as H-*ras12V*-Tg HCC model mice also showed that the expression of ATIC was enhanced in liver tumors compared to normal liver tissues (Figure 2E, Appendix A). Its expression was highly elevated in HCC large tumor (LT) from HBx-Tg mouse (*p* < 0.0001) and significantly increased in the small tumor (ST) as well as in H-*ras12V*-Tg tumors also. The expression analysis of *ATIC* by GENT (http://gent2.appex.kr/gent2/) showed a significant increase of *ATIC* expression in human liver cancers (Appendix A). GENT analysis also showed that *ATIC* expression is elevated in various extrahepatic tumors, including colon and lung cancer. TCGA database analysis on protein expression (https://www.proteinatlas.org/ENSG00000138363-ATIC/pathology) also showed the elevation of ATIC in various tumors (Appendix A). More importantly, ATIC was suggested as a poor prognosis marker of liver cancer (Appendix A). As a whole, these results indicate that ATIC overexpression is common in various tumors during tumor progression.

To induce the expression of a specific TA autoantibody, the elevated intracellular antigen must be exposed to the immune system. Because most TA antigens reported until now are intracellular proteins, such as p53, the release of cellular components following cell death or necrosis that accompanies tumorigenesis has been assumed to expose intracellular proteins to the immune system [3]. However, recent studies on tumor-associated exosomes have shown that the intracellular components included in exosomes can be exposed to immune cells even without cell disruption to stimulate or suppress immune responses [19,22,25]. ATIC is located in the cytoplasm as well as the plasma membrane [26]. Moreover, ATIC has been identified as a component of tumor-associated exosomes, including bladder, ovarian, and melanoma cancer [27,28,29]. Exosomes enriched from hepatoma cell-cultured media (HepG2 and Huh7) were resolved by Western blotting. ATIC was detected using a commercial anti-ATIC antibody and XC154 mAb in the exosomes and the cell lysate (Figure 2F). These results collectively indicate that ATIC expression is elevated during tumor progression and can be released from liver cancer tissues as a component of exosomes, which may act as an autoantigen inducing tumor-associated antibody.

### 2.3. Specific Mimotopes Against the XC154 Antibody Screened from a Phage Display Random Cyclic Heptapeptide Library Was Selected as a Detection Antigen for Serum Anti-ATIC Autoantibody

The mouse ATIC protein (NP_080471.2), which induced XC154 TA autoantibody in the HCC model mouse, shows a 91% amino acid sequence identity (95% similarity) to the human ATIC (NP_004035.2) among 592 amino acids (Appendix A). Despite such differences in amino acid sequences, the antigenic sequences of mouse ATIC and human ATIC showed very similar pattern when analyzed by the antigenicity prediction method of Kolaskar and Tongaonkar (http://imed.med.ucm.es/Tools/antigenic.pl: [30]; Appendix A). Because of the similarity in antigenic sequences, TA autoantibodies generated in tumor-model mice seem to detect the ATIC protein in human tumor cells, as shown above. Therefore, we hypothesized that human TA autoantibody against ATIC sharing the same epitope with XC154 autoantibody is generated in human HCC patients and examined whether human anti-ATIC autoantibody can be used as a biomarker of HCC.

To validate anti-ATIC autoantibody in human serum as a diagnostic marker, an appropriate capture antigen is necessary. We had expressed the recombinant human ATIC (hATIC) in the T7 shuffle *E. coli* cells, a prokaryotic expression system which forms the disulfide bonds in the cytoplasm (Appendix A). Human ATIC was expressed as a soluble protein in the T7 shuffle system, and was purified using His-affinity tag. Its reactivity against a commercial anti-ATIC antibody and XC154 mAb was shown by Western blot analysis (Appendix A). Recombinant hATIC was also detected with XC154 mAb by ELISA, although its binding affinity was relatively low (Appendix A). A competitive Western blot analysis was performed to validate whether the epitope displayed on recombinant hATIC can mimic the cellular antigenic epitope. Recombinant protein hATIC did not inhibit the binding of XC154 mAb to endogenous antigen in tumor cell lysates even at high concentration (Appendix A), which confirmed again that its affinity to XC154 mAb is relatively low. Alternatively, we screened the phage display random cyclic peptide CX_7_C library to obtain a suitable capture antigen for anti-ATIC autoantibody as in our previous studies [19,22,23,24]. We repeated the bio-panning of the M13 phage library containing 10^11^ cyclic peptides five times to isolate the specific mimotopes against XC154.

Autoantibody and amplified selected phages (Figure 3A). The cyclic peptide sequences of 10 selected phages were sequenced, and six different epitopes were obtained (Table 2). These phages were reactive to XC154 mAb but not to another TA autoantibody, XC257 IgM-type antibody (Figure 3B). The XC154 antibody-specific epitopes had a consensus sequence of C-XPSWFXR-C (Table 2). Among the selected peptide epitopes, the XC154p1 sequence (C-LPSWFHR-C) was found repeatedly (5 times per 10 phages) and is most reactive to XC154 mAb (Figure 3B, Table 2). XC154p1 M13 phage blocked the binding of XC154 mAb to its target antigen in tumor cells as shown by Western blot analysis (Figure 3C) or by FACS analysis (Figure 3D), which proves that the selected cyclic peptide with high affinity to XC154mAb can mimic the specific epitope on the natural antigen ATIC. However, such antigenic property was disappeared by linearizing the cyclic structure of the mimotope with a reducing reagent, indicating that the cyclic structure of mimotope is critical for antibody binding (Figure 3E).

Recombinant streptavidin can displace M13 phage for the display of cyclic peptide epitopes [19,22]. When the cyclic peptide epitope XC154p1 was expressed at the N-terminus of recombinant streptavidin (STA) in T7 shuffle *E. coli* cells with disulfide bond-forming activity, it maintained the specific activity to XC154 mAb (Figure 3F). XC154p1-STA also blocked the binding of XC154 mAb to endogenous antigen in tumor cells, as shown by competitive Western blot analysis (Figure 3G) as well as by FACS analysis (Figure 3H). XC154mAb showed high affinity to XC154p1-STA, compared to recombinant hATIC, as shown by ELISA (Figure 3I, Appendix A).

Collectively, tumor-associated autoantibody XC154mAb, which was generated in the HCC model mouse by an endogenous immune response against tumor-associated antigen ATIC, reacted with the XC154p1 cyclic peptide epitope with high affinity, and it can be used as a detection antigen in ELISA alternative to recombinant hATIC.

### 2.4. Human Serum Enzyme-Linked Immunosorbent assay (ELISA) to Detect Autoantibodies Against ATIC Using Specific Mimotope, XC154p1

Human serum ELISA of anti-ATIC autoantibody biomarker was performed with the optimized conditions for each step. XC154p1-STA antigen was used as a detection antigen in ELISA using the Maxisorp plate. Human sera were pretreated with albumin-depletion resin to remove albumin, which is a typical source of potentially reducing in blood [19,31], and diluted 50-fold in protein-free blocking buffer (PFBB). ELISA for human serum anti-ATIC autoantibody against 144 HCC patients and 85 healthy control subjects was performed using these conditions (Cohort 1). The anti-ATIC autoantibody biomarker level was described by the difference of OD in ELISA between XC154p1-STA and STA. As shown in Figure 4A, the reactivity of HCC patient sera against ATIC was significantly higher than that of the healthy subject sera, with an area under curve (AUC) value of 0.8755 [95% confidence interval (CI) 0.8337–0.9173, *p* < 0.0001]. The sensitivity of this ELISA was 70.83%, and the specificity was 90.63% for the cutoff value of 0.2234. Serum AFP levels were also determined for the same cohort. The AFP level in HCC patient sera was significantly higher than that of the healthy control, with an area under curve (AUC) value of 0.9089 (95% CI 0.8697–0.9418, *p* < 0.0001). The sensitivity was 42.36%, and the specificity was 100% for the cutoff value of 36.6. The plot showing the percentage of positive cases confirmed the correlation of anti-ATIC response or AFP with HCC (Figure 4B). Normal subjects had shown a low AFP level under the cutoff value; however, only 42% of HCC patients showed AFP level above the cutoff value. On the contrary, the positive detection of anti-ATIC autoantibody with XC154p1-STA was up to 65% in HCC patients. Some of the normal subjects (9%) also showed a positive response to anti-ATIC autoantibody.

We also analyzed the anti-ATIC autoantibody and AFP, according to tumor-node-metastasis (TNM) staging, tumor size, and viral infection (Figure 4C, Table 3). Anti-ATIC antibodies were detected in patients’ sera even at early tumor stages (TNM stage 1) and increased gradually to TNM stage 4. Anti-ATIC antibodies were detected in patients with a small size of the tumor (T < 2 cm) but decreased in patients with large size tumor (T > 5 cm). Interestingly, viral infection, a significant cause of HCC, did not influence the autoantibody levels. HCC patients without viral infection showed a high level of autoantibody response (~80%) as in patients with viral infection (~60%). Another cohort composed of patients with chronic hepatitis, cirrhosis and HCC (Cohort 2) showed the anti-ATIC response (Figure 4D), although its level is lower than that of HCC. Depending on the tumor stage and size, AFP-positive patients (>40 ng/mL) were increased although at a low level (20~40%). Contrary to anti-ATIC response, HCC patients without viral infection showed low AFP levels (10% compared to 80%; Figure 4D). AFP tests for cohort 2 showed a gradual increase of AFP in human serum from chronic hepatitis to cirrhosis and HCC, with a similar increase in anti-ATIC response (Figure 4D).

### 2.5. The Simultaneous Detection of AFP and Anti-ATIC Autoantibody in Patient Serum Improved HCC Diagnosis Accuracy

The correlations between anti-ATIC response and AFP were examined by Pearson analysis for two cohorts (Figure 5A). The correlations between the two biomarkers were very low (Pearson coefficient r = 0.08659 for cohort 1, 0.1072 for cohort 2). It implicates that unrelated mechanisms regulate the appearance of these two serum markers, and the combinational detection of these two markers may enhance the diagnostic efficiency of HCC. Anti-ATIC autoantibody and AFP biomarkers were indicated in different ranges of detection values (AU vs. ng/mL). To examine the effects of combined analysis of these biomarkers, the response of each biomarker was simplified to positive or negative and scored as 1 or 0 depending on whether the detection value was higher or lower than the cutoff value. The combined analysis of these two markers was performed by the simple addition of each score, which results in values as 0 (−), 1 (+), and 2 (++) (Figure 5B). ROC curve analysis was also performed for these new scores (Figure 5C). AFP biomarker showed 42.3% sensitivity, 100% specificity, and 68.32% accuracy. Anti-ATIC autoantibody showed more enhanced sensitivity and accuracy (62.38%, 74.80%), and 93.28% specificity than AFP. The combined analysis of these two markers increased the AUC value from 0.7~0.8 to 0.9. The accuracy was also increased from 68.32% to 86.64% (Figure 5C). In conclusion, the combination analysis of two independent biomarkers, anti-ATIC autoantibody and AFP, can increase the efficiency of HCC diagnosis.

## 3. Discussion

Here, we have identified anti-ATIC autoantibody as a novel tumor-associated autoantibody biomarker in HCC patients. To explain its characteristics as a TA autoantigen, we confirmed the up-regulation of ATIC in tumor model mice and human tumor cells. We also investigated its secretion from HCC cells as an exosomal component and suggested that TA exosome enriched with ATIC elicit the antigen-specific immune responses. We had screened the mimicry of the conformational epitope of anti-ATIC autoantibody from a random cyclic peptide library and successfully measured the serum anti-ATIC autoantibody by ELISA using the epitope mimicry-display protein as a coating antigen. Additionally, we tested anti-ATIC-autoantibody biomarker and conventional biomarker AFP simultaneously in the patient’s sera and suggested that the combinational measurement of two biomarkers generated by different mechanisms can enhance the diagnostic accuracy in cancer.

Unrestricted cell proliferation is a hallmark of cancer. Purines are essential components of nucleotides in cell proliferation; thus, aberrant purine metabolism is associated with cancer progression [32]. The *de novo* biosynthesis of purine depends on six enzymes to catalyze the conversion of phosphoribosyl pyrophosphate to inosine 5′-monophosphate. ATIC is a component of these enzymes cluster called purinosome and catalyzes the last two steps of purine biosynthesis [33]. ATIC upregulation in cancer has been shown in various tumors (Appendix A). ATIC also has been suggested as a poor prognosis marker of HCC [30]. These results are consistent with our results in this study and provide a rationale for anti-ATIC autoantibody as a TA biomarker.

Although the exact causes remain to be elucidated, autoantibody production has been explained by (1) tolerance defects and inflammation, (2) changes in expression levels or aberrant localization of tumor-associated antigens, or (3) neo-epitopes that may result from the altered protein structure arising through post-translational modification, mutations or occurrence of other forms of neo-epitopes [3,18]. Based on these assumptions, TA autoantibody screening methods have been established to display native oncogenic antigens as much as possible. Serological analysis of expression cDNA libraries (SEREX) using bacteriophages can reflect the changes in expression levels or in neo-epitopes that may result from the altered protein structure arising through mutations [34]. Serological proteome analysis (SERPA) or multiple affinity protein profiling (MAPPing) utilize ‘native’ proteins from tumor cells for the screening of TA autoantibodies, thereby to maintain potential protein post-translational modifications which may influence antigenicity [35,36]; however, technical difficulties of these methods limit their use.

When the target antigen of TA autoantibody was identified as ATIC, we tried to use the recombinant human ATIC in serum autoantibody ELISA as a coating antigen. Recombinant hATIC, which was prepared as a soluble protein in the *E. coli* expression system, showed specific binding to XC154 mAb and commercial anti-ATIC antibody, as shown by ELISA and Western blotting. However, its affinity was low, and it could not compete with cellular ATIC for the binding to anti-ATIC autoantibody. Instead of using cell lysates or expression libraries of tumor cells, we directly screened an epitope-mimicry of autoantibody. In contrast to hATIC, the epitope mimicry of XC154 anti-ATIC autoantibody showed a high affinity to autoantibody. It effectively blocked the autoantibody binding to cellular ATIC, and successfully detected serum autoantibody in ELISA. From these results, we assumed that the XC154 anti-ATIC autoantibody reacts with a conformational or neo-epitope modified from its sequential epitope; the recombinant hATIC expressed in *E coli* has the same amino acid sequence as ATIC expressed in human cancer cells, but may not be sufficient to form a tertiary structure of ATIC due to differences in the processes involved in protein folding. In addition, the secondary modifications (phosphorylation, acetylation, etc.) associated with the protein function in human cells are not performed in *E. coli*, which may cause some differences in the structure of epitopes to TA autoantibodies. In fact, the post-translational modification of ATIC protein in human cells was confirmed in about 80 cases (https://www.phosphosite.org/protein). These secondary modifications can affect the tertiary structure of the ATIC protein and consequently confer the neo-antigenic structure.

In addition to the neo-epitope structure, the exosomal secretion of oncogenic ATIC into extracellular space, which corresponds to aberrant localization or external exposure of overexpressed TA antigens, seems to be another important aspect for the autoimmune response. Exosomes are extracellular vesicles (EVs) that are cell-derived membranous structures [37]. The secretion of EVs was initially described as a method for eliminating unneeded compounds from cells [38]. However, exosomes have been shown to act as signaling vehicles for cell-to-cell communication because cells exchange bioactive components, including nucleic acids and proteins, via exosomes through delivery into the cytoplasm of recipient cells. Oncosomes are atypically large (1–10 µm diameter) cancer-derived EVs [39]. They contain abundant molecules and are associated with advanced disease. In our previous studies, the autoantigens, including EIF3A and SF3B1, have been shown as components of exosomes [19,22]. Capello et al. also showed that exosomes harbor B cell targets in cancer and patient plasma reacts to the tumor-cell derived exosomes [25]. Serum anti-ATIC-autoantibody was increased in HCC patients, and it also elevated in HCC-related liver diseases, including cirrhosis and chronic hepatitis. Considering our hypothesis that secreted exosomal antigens can stimulate the generation of TA autoantibody [19,22], the existence of TA autoantibody in cirrhosis or CHB implicates that certain changes in cellular physiology also occur in patients with cirrhosis or CHB, which promote the secretion of disease-associated exosomes. We expect that further studies on exosomes related to liver cells in heath and disease may explain the mechanism of autoantibody generation and its meaning as an early biomarker.

## 4. Materials and Methods

### 4.1. Tumor-Associated Monoclonal Autoantibody, XC154

We selected B-cell hybridoma clones that secreted a monoclonal TA autoantibody reactive to hepatoma cells as described previously [19] and XC154, one of TA autoantibody from them was purified and characterized. The isotype of XC154 autoantibody was determined as IgM using an isotyping kit (Thermo Fisher Scientific, Waltham, MA, USA). XC154 monoclonal autoantibody was purified from hybridoma cell culture media using protein L agarose (Thermo Fisher) and analyzed by SDS-PAGE and Western blotting.

### 4.2. Tumor Cell Lines and Serum Samples

The human cancer cell lines (HepG2, Hep3B, PLC/PRF5, Huh7, SK-Hep1, SNU638, A549, HT29, LNCaP/LN3, HeLa, SK-BR3) were obtained from the American Type Culture Collection (Manassas, VA, USA) and cultured in Dulbecco’s modified Eagle’s medium or RPMI-1640 (Invitrogen, Waltham, MA, USA) supplemented with 10% fetal bovine serum (FBS; Sigma-Aldrich, St. Louis, MO, USA). Conditioned media containing exosomes were prepared from 70–80% confluent cells, which were cultured for 48 h with exosome-free FBS (System Biosciences, Palo Alto, CA, USA). Human HCC serum samples of cohort 1 were provided by the Ajou Human Bio-Resource Bank (AHBB: Suwon, Korea), a member of the National Biobank of Korea, which is supported by the Ministry of Health and Welfare (MOHW). All samples derived from the National Biobank of Korea were obtained with informed consent and using institutional review board (IRB)-approved protocols. The study was also approved by the Public Institutional Bioethics Committee designated by MOHW (P01-201409-BS-03; Republic of Korea). Human sera from patients with HCC-related diseases in cohort study 2 were provided by the Pusan National University Yangsan Hospital (Yangsan, Korea) with patient’s consent. The acquisition of samples was reviewed and approved by the Institutional Review Board (KRIBB-IRB-20110808-04). Normal human serum samples collected at the Korean Red Cross were exempted from IRB approval, which was confirmed by Research Blood Examination Committee. Serum samples were kept at −70 °C until use.

### 4.3. Western Blot Analysis and Immunoprecipitation

The whole cell or tissue lysates were prepared using radioimmunoprecipitation assay (RIPA) buffer, as previously described [19]. Exosomes were collected from the conditioned media using exosome isolation kit (System Biosciences) or by ultracentrifugation [40] and solubilized with RIPA buffer. The protein concentration was determined by the Bradford method (Bio-Rad Laboratories, Hercules, CA, USA), and the indicated amounts of protein were analyzed by Western blotting with XC154 or anti-ATIC antibody (Thermo Fisher: MA1-086); β-actin or GAPDH were probed as a loading control. ALIX was probed as an exosome marker and calnexin was proved as cell contamination marker. Antibodies used in this study were as following: anti-calnexin antibody (Santa Cruz Biotechnology, Dallas, TX, USA: SC46669), anti-ALIX antibody (Merck Millipore, Burlington, MA, USA: #ABC40), anti-GAPDH antibody (Santa Cruz Biotechnology: SC47724), anti-β-actin antibody (Santa Cruz Biotechnology: SC8432), anti-HIS_6_ antibody (Qiagen, Hilden, Germany: #34660). Anti-mouse IgG-HRP (Cell Signaling Technology Danvers, MA, USA: #7076S) was used as secondary reagent. Band intensities were quantified using Image J (NIH, USA) and the relative intensity compared to β-actin or GAPDH was calculated.

For the competitive Western blot assay, 5 μg of the primary antibody XC154 in 10 mL of 5% skim milk solution in TBS was pre-incubated with the recombinant ATIC protein (7 μg), epitope-displaying phages (10^12^ pfu), or epitope-displaying streptavidin (2 μg) for 90 min. For the immunoprecipitation assay, HepG2 cell lysates (1 mg) were incubated with 8 μg of anti-ATIC antibody and protein A/G beads or XC154 antibody and protein L beads (Santa Cruz Biotechnology) for 4 h at 4 °C. After briefly washing with phosphate-buffered saline (PBS), the immunoprecipitates were analyzed by Western blotting. As isotype control antibody, anti-GAPDH antibody (Mouse IgG, Santa Cruz Biotechnology: SC47724) and anti-FOXO2 antibody (Mouse IgM, Santa Cruz Biotechnology: SC393873) were used.

### 4.4. Immunofluorescence

To confirm the cellular localization of target antigen against XC154 mAb, tumor cells were fixed and permeabilized with BD cytofix/cytoperm solution (BD Bioscience, Franklin Lakes, NJ, USA: 554714). Then, cells were washed with 2 mL of BD cytoperm/wash solution (BD: 554714) twice and incubated with XC154 mAb in cytoperm/wash solution (5 μg/mL) at 4 °C for 1 h. After washing as above, cells were treated with goat anti-mouse IgG-FITC (Abcam, Cambridge, UK: ab6785) in BD cytoperm/wash solution (1:1000) at 4 °C for 1 h. The stained cells were analyzed by Zeiss LSM510 Meta microscope (Carl Zeiss MicroImaging, Inc., Thornwood, NY, USA).

### 4.5. Immunohistochemistry

Preparation of paraffin-embedded tissue specimens from the tumor-model mouse and immunostaining with anti-ATIC antibody (Thermo Fisher, 6 μg/mL) or XC154 mAb (5 μg/mL) were performed following procedures described as previously [14]. Paraffin-embedded human HCC AccuMax Array was purchased from ISU Abxis Co (Seoul, Korea). The photomicrographs were acquired at 200× or 400× magnification. The intensity of each staining was quantified by Image J and blotted.

### 4.6. Identification of XC154 Target Antigen

For the enrichment of the antigen against XC154 autoantibody, LNCap-LN3 cell lysates were prepared with RIPA cell lysis buffer and fractionated using Hitrap-Q HP ion exchange columns (GE, Boston, MA, USA). Fractionation was performed with a linear gradient from 0 to 1 M NaCl dissolved in PBS (pH 7.4), and positive fractions containing the XC154 antigen were confirmed by Western blot analysis. Selected fractions containing XC154 antigen were pooled, concentrated, and separated by 10% SDS-PAGE, followed by Western blotting or Coomassie blue staining. The stained band that corresponded to protein band reactive to XC154 antibody was excised and used for in-gel digestion. Protein identification using the in-gel protein digest was performed with nano-liquid chromatography-electrospray ionization-tandem mass spectrometry (LC/ESI-MS/MS) and Mascot data base search, as described previously [24].

### 4.7. Knockdown of ATIC and Reverse Transcription-Polymerase Chain Reaction (RT-PCR)

To confirm that the XC154 antigen as ATIC, HepG2 cells were transfected with siRNA targeting ATIC (Bioneer Corporation, Daejeon, Korea) using Lipofectamine RNAimax reagent (Thermo Fisher). The sequences of siRNA were as follows; si-ATIC-1 sense: 5′-GUC UCG UAU ACA CUG CAC U(dTdT)-3′, antisense: 5′-AGU GCA GUG UAU ACG AGA C(dTdT)-3′; si-ATIC-2 sense: 5′-CUC UGA GUU GAC GGG AUU U(dTdT)-3′, antisense: 5′-AAA UCC CGU CAA CUC AGA G(dTdT)-3′; si-ATIC-3 sense: 5′-GGA GGC UUU AGG UAU UCC A(dTdT)-3′, antisense: 5′-UGG AAU ACC UAA AGC CUC C(dTdT)-3′. ATIC-knockdown cells were analyzed 72 h after transfection. RT-PCR was performed using primer pairs purchased from Bioneer, as follows; ATIC forward: 5′-AAC CAG AGG ACT ATG TGG TG-3′, reverse: 5′-TCT TGG CGT AGC ACA CAG AG-3′; β-actin: 5′-AAT CTG GCA CCA CAC CTT CTAC-3′, reverse: 5′-ATA CGA CAG CCT GGA TAG CAAC-3′.

### 4.8. Bio-Panning of the XC154-Specific Cyclic Peptide Mimotopes

For the selection of the mimotope specific to XC154, the phage display random cyclic peptide library Ph.D.-C7C™ (New England Biolabs, MA, USA) was used. Panning was repeated five times, and sequencing of selected mimotope phages was performed according to the manufacturer’s instructions.

### 4.9. ELISA

ELISA was performed as described previously [19]. In brief, ELISA plate (Maxisorp; Nunc, Rochester, NY, USA) was coated with indicated amount of antigen (recombinant hATIC, cyclic peptide epitope-display M13 phages or epitope-fused STA) in PBS (pH 7.4) overnight at 4 °C, and blocked with Protein-Free-Blocking Buffer (PFBB; Thermo Fisher). Primary antibody XC154 solution was also prepared in PFBB, and anti-mouse IgGAM-HRP (Thermo Fisher) was used as secondary antibody. To evaluate the effect of disulfide bonds on the antigenicity, cyclic peptide display phages were linearized by reduction and alkylation as described previously [24] and used as coating antigens. For the detection of human autoantibody against ATIC in patient sera, ELISA plates were coated with XC154p1-STA at 500 ng/well. After blocking with PFBB, the plates were treated with albumin-depleted human sera (1:50 diluted in PFBB) for 90 min and detected by horseradish peroxidase (HRP)-conjugated anti-human IgGAM antibody (Thermo Fisher; 1:2000 diluted). Albumin depletion from human serum was performed using Affi-Gel^®^ Blue Gel (Bio-Rad), following the manufacturer’s instructions. STA without a peptide epitope (STA) was used as control coating antigen. Serum AFP was quantified using commercial kit (R&D Systems, Inc. Minneapolis, MN, USA).

### 4.10. Competitive FACS

For the flow cytometric analysis, the suspended cells were fixed and permeabilized with BD cytoperm/cytofix solution (BD), followed by the incubation with primary antibody solution and with goat anti-mouse IgG F(ab′)_2_-PE. The stained cells were analyzed by FACScalibur (BD) and the obtained data were analyzed using CellQuest software (BD). When determining whether the autoantibody-mimotope display phage or streptavidin can compete with target cellular antigen for antibody binding, primary antibodies were pre-incubated with each mimotope-display antigens at room temperature for 60 min and then used for staining.

### 4.11. Expression and Purification of Recombinant ATIC Protein and XC154p1 Epitope Display Streptavidin (XC154p1-STA)

The cDNA of human ATIC (hATIC: 1779 bp) was prepared from total mRNA of HepG2 cells, and amplified cDNA insert was cloned into pET28a(+) vector using NheI and BamHI sites (Appendix A). For the effective protein folding of hATIC which containing 10 cysteine residues, hATIC expression vector was transformed into *E. coli* strain SHuffle^®^ T7 (New England Biolabs), an engineered *E. coli* B strain, as a host cell to promote disulfide bond formation in the cytoplasm. Transformed cells were cultured for 22 h at 25 °C and His_6_-tagged hATIC was purified using a Talon affinity column (Clontech Laboratories, Mountain View, CA, USA) by imidazole gradient elution (0 to 0.5 M). Fractions containing hATIC were pooled and used for further analysis. The DNA coding cyclic peptide epitope XC154p1 sequence (-CLPSWFHRC-) with NdeI and NotI restriction enzyme sites was synthesized (Bioneer) and cloned into the streptavidin cloned-pET28a(+) vector, as described previously [22]. The cyclic peptide epitope expression vector was also transformed into *E. coli* strain SHuffle^®^ T7. Transformants were cultured and XC154p1-STA was purified using a Talon affinity column and HiLoad 16/600 Superdex 200 prep grade column, as described previously [22]. Fractions containing XC154p1-STA were pooled and used for further analysis and human serum ELISA.

### 4.12. Statistical Analysis

All data are presented as the mean ± standard deviation (SD). A two-tailed Student’s *t*-test was used to evaluate significance. ELISA results were evaluated with an ROC curve using Prism 7 software (GraphPad Software, La Jolla, CA, USA).

## Figures and Tables

**Figure 1 ijms-21-09718-f001:**
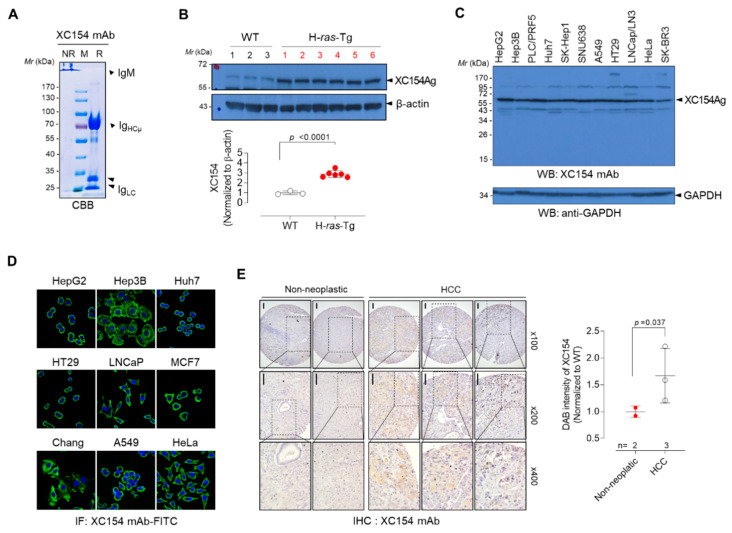
Tumor-associated autoantibody XC154mAb was identified in human (HCC) model HBx-Tg mouse. (**A**) SDS-PAGE analysis of purified XC154 mAb. Purified XC154 mAb (10 μg) was treated with non-reducing (NR) or reducing (R) SDS-PAGE sample buffer and separated on 10% SDS-PAGE gel. Coomassie blue stained gel showed high molecular weight IgM and μ heavy chain with molecular weight of 72 kDa. M: molecular weight marker. (**B**) The expression of XC154 Ag in liver tissues of H-*ras*12V-Tg mice. The liver tissue lysates (50 μg) of wild-type mice (*n* = 3) or tumor-bearing H-*ras*12V-Tg mice (*n* = 6) were separated on 10% SDS-PAGE and Western Blots were probed with XC154 mAb. Band intensities were quantified by Image J software and the values were normalized to β-actin. (**C**) Expression of XC154 antigen in various human tumor cell lines (cell lysates 40 μg) shown by Western blotting. GAPDH was served as an internal control. Arrows indicate the XC154 antigen. (**D**) Immunofluorescent staining of tumor cell lines with XC154 mAb (0.5 μg/mL) and FITC-labeled anti-mouse IgG. (**E**) Immunohistochemical staining of human liver tissues (non-neoplatic or HCC tissue) microarray with XC154 mAb (0.5 μg/mL). DAB intensities were quantified by Image J software and the relative values were plotted. Statistical significance was determined by two-tailed Student’s *t*-test.

**Figure 2 ijms-21-09718-f002:**
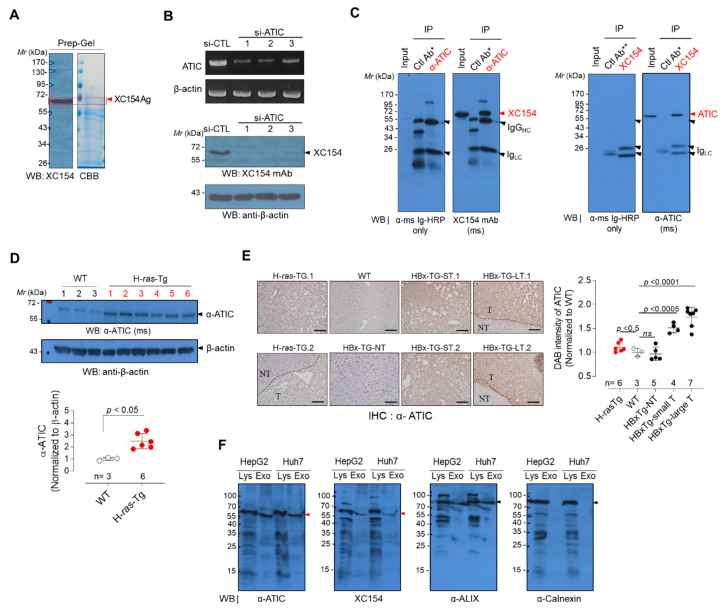
The target antigen of XC154 mAb was oncogenic ATIC. (**A**) Preparative 10% SDS-PAGE to isolate XC154 antigen and in-gel digestion for the mass spectrometric protein identification. The protein band containing XC154 antigen, which was confirmed by Western blotting, was excised (indicated by the red arrow) and in-gel digested. Proteins identified by mass spectrometric analysis were listed in Table 1. (**B**) The validation of XC154 antigen as ATIC. HepG2 cells were transfected with si-ATIC and their cell lysates were analyzed by Western blotting with XC154 mAb. β-Actin was used as an internal control. (**C**) Immunoprecipitation analysis for the verification of XC154 antigen as ATIC. Red arrows indicate XC154 Ag or ATIC. (**D**) The expression of ATIC in liver tissues of H-*ras12V*-Tg mice. Blots were probed with commercial anti-ATIC antibody. (**E**) Immunohistochemical analysis of ATIC in liver tissues of HCC model mice. Liver tissues from wild type control mice (Non-Tg: WT) were also stained. NT: non-tumor, T: tumor region, H-*ras12V*-Tg (*n* = 6), Non-Tg (*n* = 3), HBx-Tg-nonT: HBx-transgenic mouse without tumor (*n* = 5), HBx-Tg-ST: HBx-transgenic mouse with small tumor (*n* = 4), HBx-Tg-LT: HBx-transgenic mouse with large tumor (*n* = 7). Representative images were shown (All staining images were shown in Appendix A). (**F**) ATIC in exosomes purified from hepatoma cell cultured media (HepG2 and Huh7 cells; 35 μg) analyzed by Western blotting. A well-known exosomal marker, ALIX, and ER marker, Calnexin, were probed as controls. The red arrow indicates ATIC.

**Figure 3 ijms-21-09718-f003:**
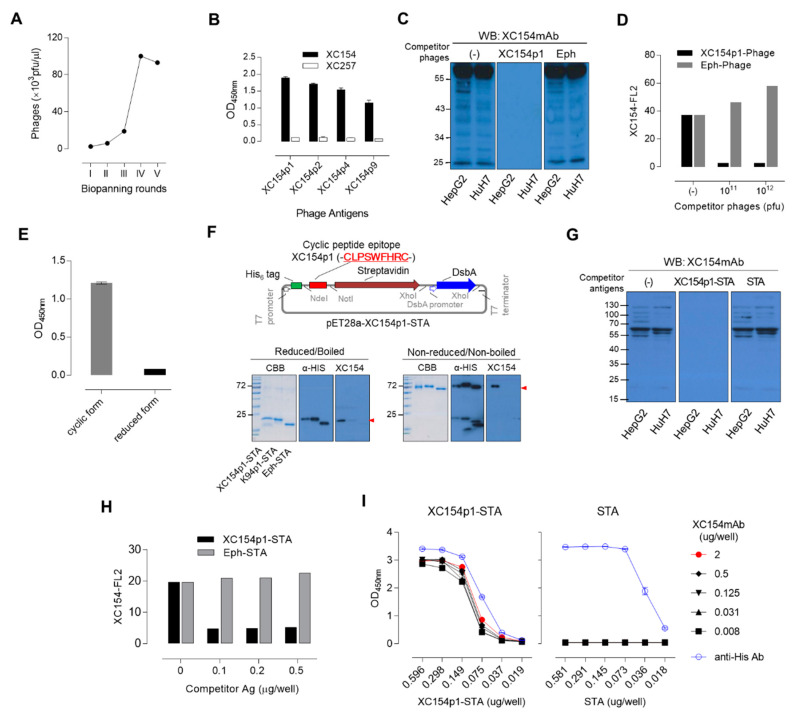
Anti-ATIC autoantibody ELISA was set up using XC154p1-STA for the specific and sensitive detection of autoantibodies in human sera. (**A**) Bio-panning with XC154 antibody against phage-displayed random cyclic heptapeptide library. (**B**) Phage ELISA to confirm the specific binding of selected mimotopes against XC154. XC257, another autoantibody was compared as a non-related control. Coating phages were 10^11^ pfu phages per well. Primary antibody XC154 mAb was used at the concentration of 0.1 μg/well. (**C**) Competitive Western blot analysis on the specific binding of XC154p1 phage to XC154 mAb. The amount of protein loading was 40 μg/lane, and XC154 mAb (5 μg/10mL) was pre-incubated with each phage (10^12^ pfu). (**D**) Competitive FACS assay of XC154p1 phage to evaluate specific binding of XC154 antibody to HepG2 cells. 1 × 10^5^ cells were reacted with XC154 mAb (0.5 μg/reaction), and competitive phages were pre-incubated with each phage as indicated in the plot. (**E**) The disappearance of XC154 antibody specific binding by linearizing XC154p1 cyclic peptide mimotope. (**F**) Expression vector of XC154p1 cyclic peptide-fused STA (XC154p1-STA). The purified XC154p1-STA antigen was analyzed by SDS-PAGE and Western blotting to confirm its tetrameric status. The monomer forms of STA antigens were confirmed by the analysis of reduced as well as boiled antigens. Purified protein loading amount 5 or 0.5 μg/lane (immunostaining with anti-His antibody (α-HIS) or XC154 autoantibody (XC154)). (**G**) Competitive Western blot analysis on the specific binding of XC154p1-STA to XC154 mAb. The cell lysates were loaded 40 μg/lane, and XC154 mAb (5 μg/10mL) was pre-incubated with XC154p1-STA or STA (2 μg). (**H**) Competitive FACS assay of XC154p1-STA to evaluate specific binding of XC154 antibody to HepG2 cells. 1 × 10^5^ cells were reacted with XC154 mAb (0.5 μg/reaction), and competitive antigens were pre-incubated with each antigen (XC154p1-STA or STA) as indicated in the plot. (**I**) XC154p1 epitope ELISA with XC154p1-STA antigen. Antigen was coated as indicated amount and detected with gradually diluted XC154 antibody.

**Figure 4 ijms-21-09718-f004:**
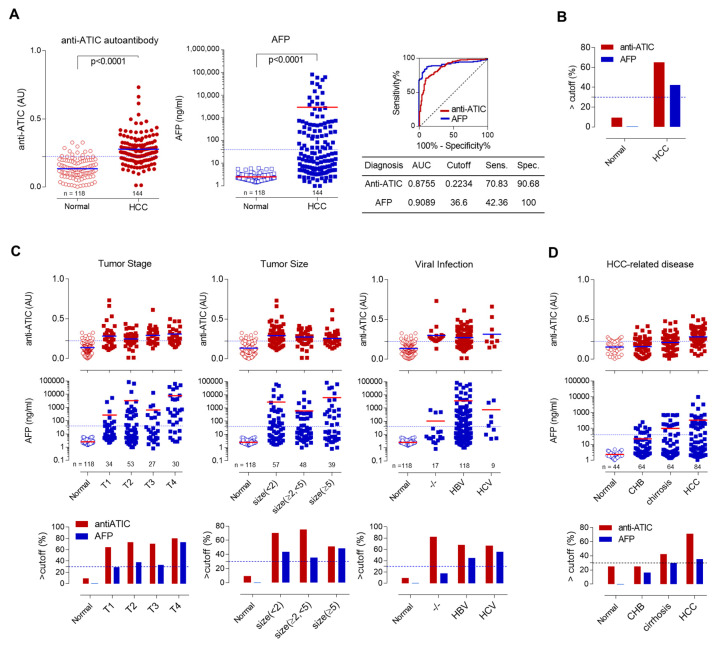
Human serum anti-ATIC ELISA using XC154p1-STA discriminated HCC patients from normal subjects. (**A**) Anti-ATIC autoantibody ELISA using XC154p1-STA in HCC patient’s sera as well as normal subjects. The specific binding of serum autoantibody to XC154p1 epitope (anti-ATIC response) was described as the difference in OD values between XC154p1-STA reaction and that of STA reaction. AFP was measured also for the same cohort. Normal (*n* = 118), HCC (*n* = 144). ROC curve analysis showed the diagnostic efficiency of each biomarker. All experiments were performed in duplicate and repeated a least three times. (**B**) The percentage of subjects with biomarker response over the cutoff value. (**C**) Anti-ATIC response related to tumor stage, tumor size or viral infection. The top panels show the anti-XC154p1 response of each sample. The lower panels show the percentages of each group over the cutoff value (CV). The level of AFP depending on these factors are also shown. (**D**) Anti-ATIC autoantibody ELISA using XC154p1-STA in cohort 2 consists of normal (*n* = 42), chronic hepatitis (*n* = 64), cirrhosis (*n* = 64) and HCC (*n* = 84). AFP levels of this cohort were also compared. The clinicopathological features, including the above results, are described in detail in Table 3.

**Figure 5 ijms-21-09718-f005:**
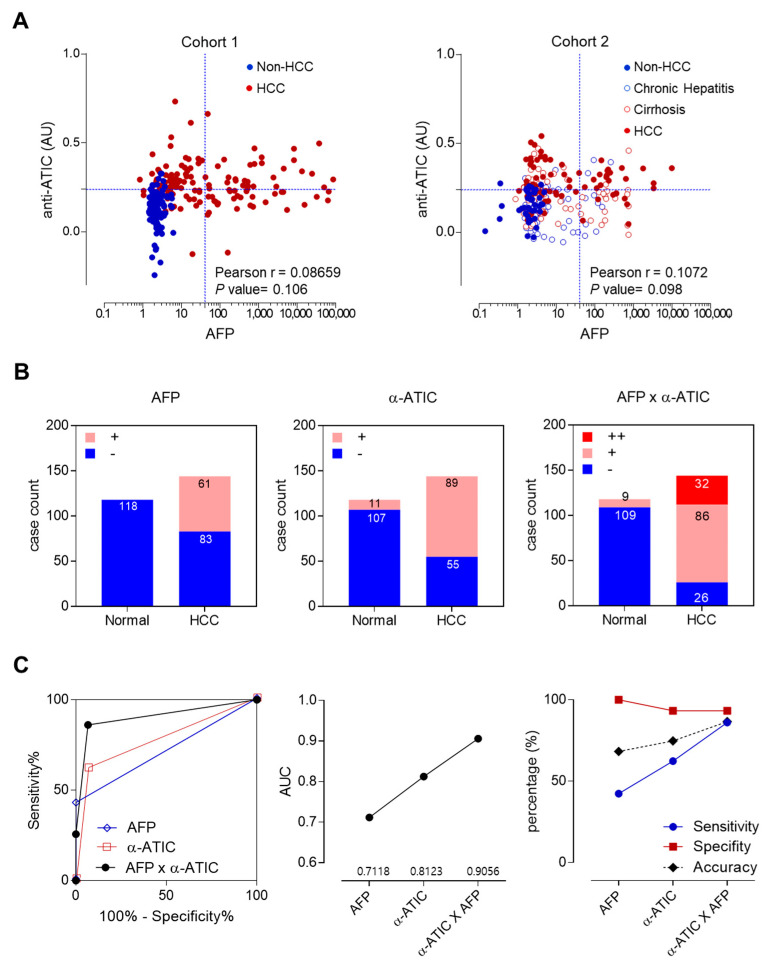
Combinational analysis of HCC biomarkers enhanced the diagnostic accuracy. (**A**) Pearson analysis of correlations between anti-ATIC autoantibody biomarker and AFP was performed (see also Appendix A). (**B**) Combinational analysis of HCC biomarkers, anti-ATIC autoantibody and AFP. The diagnostic values of each biomarker shown in Figure 4A (anti-ATIC and AFP) were simplified as either responsive (+) or non-responsive (−) according to whether their detection values were above or below the cutoff value. Then, we analyzed the diagnostic values of each biomarker or their combination: for the combined analysis of these markers, we simply added the unified diagnostic indexes of a serum sample and designated double negative as −, single positive as +, and double positive as ++. Numbers on plots represent the number of corresponding subjects. (**C**) ROC curve analysis of combined biomarker tests with unified index shown in Figure 5B (left panel). AUC values of each ROC curve analysis were plotted (middle). Sensitivity, specificity and accuracy of each test were also plotted (right panel). The additional information of statistical analysis was shown in Appendix A.

**Table 1 ijms-21-09718-t001:** Mass spectrometric analysis of XC154 antigen.

Gene Symbol	Protein Name	Accession Number	Molecular Mass (Da)	Protein Score	No. of Matched Peptide	Sequences	emPAI *
ATIC	Bifunctional Purine Biosynthesis Protein PURH	IPI00289499	64575	1780	85 (48)	38 (26)	16.34
GLA	Alpha-galactosidase A	IPI00025869	48735	267	10 (6)	3 (3)	0.56
KRT1	Keratin, type II cytoskeletal 1	IPI00220327	65999	234	13 (10)	13 (10)	0.83
AIFM1	Isoform 1 of Apoptosis-inducing factor 1, mitochondrial	IPI00000690	66859	216	16 (6)	14 (6)	0.81
KRT9	Keratin, type I cytoskeletal 9	IPI00019359	62027	206	11 (7)	9 (7)	0.69
TARS	Threonyl-tRNA synthetase, cytoplasmic	IPI00329633	83382	197	12 (6)	12 (6)	0.42
HADHA	Trifunctional enzyme subunit alpha, mitochondrial	IPI00031522	82947	168	5 (4)	5 (4)	0.19
ENO1	Isoform alpha-enolase of Alpha-enolase	IPI00465248	47139	113	3 (3)	2 (2)	0.26
FH	Isoform Mitochondrial of Fumarate hydratase, mitochondrial	IPI00296053	54602	106	2 (1)	2 (1)	0.07
CS	Citrate synthase, mitochondrial	IPI00025366	51680	97	5 (2)	3 (1)	0.23

* emPAI: exponentially modified abundance index.

**Table 2 ijms-21-09718-t002:** Epitope peptide sequences (C-X_7_-C *) of selected phages.

Phage Antigens	Epitope Amino Acid Sequences (X_7_)							Frequency **
XC154p1	L ***	P	S	W	F	H	R	5/10
XC154p2	A	P	S	W	L	H	R	1/10
XC154p3	D	P	S	G	H	R	A	1/10
XC154p4	S	P	S	G	L	F	S	1/10
XC154p8	A	P	S	W	F	F	R	1/10
XC154p9	T	P	S	W	F	T	R	1/10

* C: cysteine, X: amino acids except cysteine; ** Frequency: the cyclic peptide sequences of 10 selected phages were sequenced and the frequency of each mimotope was indicated as n/10; *** abbreviation of amino acids.

**Table 3 ijms-21-09718-t003:** Patient details in validation cohorts *.

	Cohort 1 *	Cohort 2 **		
Patients (number)	HCC	CHB	LC	HCC
Gender				
male/female,	114/30	43/21	40/24	56/17
*n* (%)	(79.2/20.8)	(67.2/32.8)	(62.5/37.5)	(76.7/23.3)
Age distribution, yr (Avg ± SD)	33~83 (55.4 ± 10.3)	23~67 (49.2 ± 10.1)	41~78 (54.2 ± 9.08)	41~80 (58.2 ± 7.69)
Serum AFP (ng/mL)	0.8~83000	1.4~252	0.8~718.9	1.08~9786.3
<CV_40_/>CV_40_	83/61	54/10	44/20	47/26
*n* (%)	(57.6/42.4)	(84.4/15.6)	(68.8/31.2)	(64.4/35.6)
Viral infection				
(–)/HBV/HCV	17/118/9	0/64/0	4/60/0	15/58/0
*n* (%)	(11.8/81.9/6.3)	(0/100/0)	(6.2/93.8/0)	(20.5/79.5/0)
Tumor stage T1/T2/T3/T4 *n* (%)	34/53/27/30 (23.6/36.8/18.6/20.8)	n.a. ^§^	n.a.	n.a.
Tumor size (cm) T < 2/2 < T < 5/5 < T *n* (%)	57/48/39 (39.6/33.3/27.1)	n.a.	n.a.	n.a.
Anti-ATIC autoantibody (AU)	0.01~1.1	0~0.416	0~0.479	0.043~0.539
<CV_0.223_/>CV_0.223_,	55/89	48/16	38/26	26/47
*n* (%)	(38.2/61.8)	(75/25)	(59.4/40.6)	(35.6/64.4)

* Human HCC serum and normal serum samples for this study were provided by the Ajou Human Bio-Resource Bank (AHBB), a member of the National Biobank of Korea. All samples derived from the National Biobank of Korea were obtained with informed consent under institutional review board-approved protocols and the study was approved by Public Institutional Bioethics Committee designated by MOHW ((P01-201409-BS-03; Republic of Korea). In cohort study 2, human sera of HCC-related diseases were provided by the Pusan National University Yangsan Hospital, Korea with patient’s consent. The acquisition of samples was reviewed and approved by the Institutional Review Board (KRIBB-IRB-20110808-04). Normal human serum samples collected at the Korean Red Cross were exemption from IRB approval which was confirmed by Research Blood Examination Committee, and parts of them were used for the study of cohort 1 (*n* = 118) and cohort 2 (*n* = 42). The ratio of gender portion is 68/32 (male/female), and the age distribution ranges from 24 to 66 (Avg ± SD: 40.9 ± 11.8). ** § n.a.: not available. Abbreviations: CHB: chronic hepatitis B; LC: liver cirrhosis; HCC: hepatocellular carcinoma.

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
