# Peer review of "Cyclic Peptide Mimotopes for the Detection of Serum Anti–ATIC Autoantibody Biomarker in Hepato-Cellular Carcinoma"

_ijms, 2020, doi:10.3390/ijms21249718_

Round 1

Reviewer 1 Report

The authors identified TA autoantibody in HBx-transgenic hepatocellular carcinoma model mouse and found ATIC, the target antigen. It was confirmed that ATIC was upregulated in HBx-tg mouse and human HCC tissues, and was secreted through cancer-derived exosomes to trigger an immune response. The authors suggested that anti-ATIC autoantibody could be a serum biomarker related to HCC. It is a good attempt to improve the biomarker's efficiency for autoantibody using antigenic mimicry.

However, I have several minor comments as detailed below.

  • It is not clear that non-HCC and normal are different or the same in the two cohorts proposed by authors. If it was the same, authors should be better to use the same terminology.
  • Clinical information about normal patients of each cohort is missed. The author should add these details in table 3.
  • The author should represent the population number of each group in every graph in figure 4.
  • In figure 5C, additional statistical analysis to show the statistical significance of each group is needed and p-values should be included. Overall, it would be good to describe the result in more detail.

Author Response

Comments and Suggestions for Authors

Reviewer #1

  • It is not clear that non-HCC and normal are different or the same in the two cohorts proposed by authors. If it was the same, authors should be better to use the same terminology.
  • Two groups of normal serum were obtained from Korean Red Cross. Normal serum of group 1 (n=118) was used in the study of cohort 1 and normal group 2 (n=42) was used in the study of cohort 2. The terminology “non-HCC” was corrected to “Normal” (line 288, Fig. 4).
  • Clinical information about normal patients of each cohort is missed. The author should add these details in table 3.

è The clinical information of normal serum was obtained about age distribution and gender, which is inserted in Table 3. The AFP levels of normal sera were determined using AFP quantification ELISA kit in our lab.

  • The author should represent the population number of each group in every graph in figure 4.
  • The population number of each group was inserted in every graph in Figure 4 In the revised manuscript.
  • In figure 5C, additional statistical analysis to show the statistical significance of each group is needed and p-values should be included. Overall, it would be good to describe the result in more detail.

The additional statistical analysis to show the statistical significance is added to Supplementary data Fig S7.

Reviewer 2 Report

  1. Line 56. :”… despite using the highly sensible devices.”. Presumably it is meant to state, “despite using highly sensitive devices.” There a some other places where the English could be improved, although overall it is fine.
  2. Statements needing references or further explanation: Lines 58-60. “They are produced early in tumorigenesis….” This broad statement is undocumented. Even the preceding and following sentences largely use reviews as references, not primary literature. This entire section needs more rigor, either specific references or tone-down writing. In fact, the last sentence (line 65-66) seems to conflict with the preceding points.; Reference #17. This is a book chapter and is not properly referenced.
  3. Fig. 1A. This figure seems unnecessary. The monoclonal antibody was stated to be IgM using an isotyping kit. In addition, the use in the W.B. of an anti-IgG (which, strangely, also has light chain reactivity) rather than an anti-IgM heavy chain-specific antibody is enigmatic.
  4. Fig. 1D. The legend and M&M state that an anti-IgG detecting reagent was used; shouldn’t this be anti-IgM? Some description of these results (apparent cytoplasmic location) as well as the discordance with Fig. 1C data re e.g., A49 and HeLa cells would be useful.
  5. Fig. S3. The tissue expression analysis of ATIC in the GENT database was not accessible using the hyperlink, and several published papers clarifying the use of the GENT data base (Cancer Informatics 2011:10 149–157; Park et al. BMC Medical Genomics 2019, 12(Suppl 5):101) did not included the ATIC oncogene product. Better explanation of the source of the tissue expression information used in this figure would be helpful.
  6. Lines 208-209, 262-263, and Fig. S5C. The XC154 mAb dose response seems to show no binding, not “relatively low” or “compared to recombinant hATIC”.
  7. Concerning “Human sera were pretreated with albumin-depletion resin to remove albumin, which is a typical source of potentially reducing in blood [19,30] …”. What is the evidence that this awkward and impractical procedure is necessary, as opposed to the results in Fig. 3C in which the cyclic peptide was presumably reduced by boiling (with mercaptoethanol?), a much more rigorous chemical reduction process than the mere co-presence of albumin could mediate.
  8. Lines 404-405. “…although the recombinant maintained its tertiary protein structure based on its cognate sequence…” This statement is presumptuous and simplistic.
  9. Lines 424-425. “…Considering that secreted exosomal antigens stimulate TA autoantibody generation …” This undocumented and unsubstantiated statement should be revised.

Author Response

Answers to Comments or Suggestions for Authors

# reviewer 2

1. Line 56. :”… despite using the highly sensible devices.”. Presumably it is meant to state, “despite using highly sensitive devices.” There a some other places where the English could be improved, although overall it is fine.

Answer>  The statement in line 56 has been revised as the reviewer suggested. :” despite using the highly sensible devices.” --> “despite using highly sensitive devices.”

2. Statements needing references or further explanation: Lines 58-60. “They are produced early in tumorigenesis….” This broad statement is undocumented. Even the preceding and following sentences largely use reviews as references, not primary literature. This entire section needs more rigor, either specific references or tone-down writing. In fact, the last sentence (line 65-66) seems to conflict with the preceding points.; Reference #17. This is a book chapter and is not properly referenced.

Answer> The statement in lines 58-60 has been deleted because the following sentence about anti-p53 autoantibody biomarker can explain the phenomenon of the early production of autoantibody in cancer.

In the sentence of line 65-66, we have described the limitation of the detector antigen of autoantibody biomarker: In most studies on autoantibody biomarker, the recombinant protein or peptide was used as a detection antigen of TA autoantibody, which may be not sufficient to display neo-epitope of autoantigens.

Reference #17 was corrected as follows: 17. Royahem, J.; Conrad, K.; Frey, M.; Melhom, J.; Frank, K.H. Autoantibodies Predictive Parameters of Tumor Development. In Pathogenic and Diagnostic Relevance of Autoantibodies; Konrad, K., Humboldt, R.L., Meurer, M., Shoenfeld, Y., Tan, E.M. Eds..; Pabst, Berlin, Germany, 1988; pp412– 414.  

3. Fig. 1A. This figure seems unnecessary. The monoclonal antibody was stated to be IgM using an isotyping kit. In addition, the use in the WB of an anti-IgG (which, strangely, also has light chain reactivity) rather than an anti-IgM heavy chain-specific antibody is enigmatic.

Answer> As reviewer suggested, the western blot image was deleted from Fig. 1A in the revised manuscript.

4. Fig. 1D. The legend and M&M state that an anti-IgG detecting reagent was used; shouldn’t this be anti-IgM?

Answer> We have used HRP-conjugated anti-mouse IgG for the detection of XC154 antibody because HRP-conjugated anti-mouse IgG can also detect the light chains of IgM (https://images.novusbio.com/design/secondaryHandbook.pdf ).

5. Some description of these results (apparent cytoplasmic location) as well as the discordance with Fig. 1C data re e.g., A459 and HeLa cells would be useful.

Answer>The staining of XC154 antibody was also shown in the cytoplasmic location of A549 and HeLa cells. We have adjusted the brightness of the images of A549 and HeLa in revised Fig. 1D.

6. Fig. S3. The tissue expression analysis of ATIC in the GENT database was not accessible using the hyperlink, and several published papers clarifying the use of the GENT data base (Cancer Informatics 2011:10 149–157; Park et al. BMC Medical Genomics 2019, 12(Suppl 5):101) did not included the ATIC oncogene product. Better explanation of the source of the tissue expression information used in this figure would be helpful.

Answer> GENT (Gene Expression database of Normal and Tumor tissues) is updated to GENT2. We corrected the hyperlink to GENT2 in the manuscript (line 167) and Fig. S3, and renewed the data using GENT2. GENT2 also shows that ATIC is significantly overexpressed in liver cancer.

TCGA database (https://www.proteinatlas.org/ENSG00000138363-ATIC/pathology) shows ATIC is a prognosis marker of liver cancer. Minjing Li et al also proved that ATIC is a prognosis marker of liver cancer (Minjing Li et al. Bifunctional enzyme ATIC promotes propagation of hepatocellular carcinoma by regulating AMPK-mTOR-S6 K1 signaling. Cell Commun Signal. 2017, 15, 52).

7. Lines 208-209, 262-263, and Fig. S5C. The XC154 mAb dose response seems to show no binding, not “relatively low” or “compared to recombinant hATIC”.

Answer> The response of XC154 antibody to recombinant hATIC is lower than XC154p1-STA, about 3-fold, as shown in ELISA. When recombinant ATIC was detected with XC154 antibody with sufficient amount (0.5ug/mL) in Western blotting (Fig S5a, b), hATIC was clearly detected without nonspecific detection of BSA. Line 208 and 261-262 were modified to describe these facts clearly.

8. Concerning “Human sera were pretreated with albumin-depletion resin to remove albumin, which is a typical source of potentially reducing in blood [19,30] …”. What is the evidence that this awkward and impractical procedure is necessary, as opposed to the results in Fig. 3C in which the cyclic peptide was presumably reduced by boiling (with mercaptoethanol?), a much more rigorous chemical reduction process than the mere co-presence of albumin could mediate. 

Answer> As the reviewer pointed out, the reduction of the disulfide bond in cyclic peptide display streptavidin antigen requires sufficient treatment of reducing reagents, such as DTT or beta-mercaptoethanol. The reactivity of XC154p1 cyclic peptide epitope is diminished by the reduction of the disulfide bond of cyclic peptide epitopes as shown in Fig 3E, and the reducing power of human serum albumin is also evident as described in references. We tried to reduce the effect of serum albumin on the cyclic peptide epitopes in ELSA during the long incubation time (90 minutes) by depleting albumin from serum sample.

9. Lines 404-405. “…although the recombinant maintained its tertiary protein structure based on its cognate sequence…” This statement is presumptuous and simplistic. 

Answer> We agree with the reviewer that the statement in line 404-405 is presumptuous and simplistic, and have revised the statement as follows: “The recombinant hATIC expressed in E coli has the same amino acid sequence as ATIC expressed in human cancer cells, but may not be sufficient to form a tertiary structure due to differences in the processes involved in protein folding. In addition, the secondary modifications (phosphorylation, acetylation, etc.) associated with the protein function in human cells are not performed in E. coli, which may cause some differences in the structure of epitopes to TA autoantibodies.”

10. Lines 424-425. “…Considering that secreted exosomal antigens stimulate TA autoantibody generation …” This undocumented and unsubstantiated statement should be revised.

Answer> We agree with the reviewer that the statement in line 424-425 is undocumented and unsubstantiated, and have revised the statement as follows: “Considering our hypothesis that secreted exosomal antigens can stimulate the generation of TA autoantibody [19,22], the early existence of TA autoantibody in cirrhosis or CHB implicates that certain changes in cellular physiology also occur in patients with cirrhosis or CHB, which promote the secretion of disease-associated exosomes.” (line 430-433).